# The Role of Telocytes and Telocyte-Derived Exosomes in the Development of Thoracic Aortic Aneurysm

**DOI:** 10.3390/ijms23094730

**Published:** 2022-04-25

**Authors:** Thomas Aschacher, Olivia Aschacher, Katy Schmidt, Florian K. Enzmann, Eva Eichmair, Bernhard Winkler, Zsuzsanna Arnold, Felix Nagel, Bruno K. Podesser, Andreas Mitterbauer, Barbara Messner, Martin Grabenwöger, Günther Laufer, Marek P. Ehrlich, Michael Bergmann

**Affiliations:** 1Department of Cardiovascular Surgery, Clinic Floridsdorf and Karl Landsteiner Institute for Cardio-Vascular Research, 1210 Vienna, Austria; bernhard.winkler@gesundheitsverbund.at (B.W.); zsuzsanna.arnold@gesundheitsverbund.at (Z.A.); martin.grabenwoeger@gesundheitsverbund.at (M.G.); 2Department of Cardiac Surgery, Medical University of Vienna, 1090 Vienna, Austria; eva.eichmair@meduniwien.ac.at (E.E.); barbara.messner@meduniwien.ac.at (B.M.); guenther.laufer@meduniwien.ac.at (G.L.); marek.ehrlich@meduniwien.ac.at (M.P.E.); 3Department of Plastic, Reconstructive and Aesthetic Surgery, Medical University of Vienna, 1090 Vienna, Austria; olivia.aschacher@meduniwien.ac.at; 4Center for Anatomy and Cell Biology, Medical University of Vienna, 1090 Vienna, Austria; katy.schmidt@meduniwien.ac.at; 5Department of Vascular Surgery, Medical University of Innsbruck, 6020 Innsbruck, Austria; fkenzmann@gmail.com; 6Department of Biomedical Research, Medical University of Vienna, 1090 Vienna, Austria; felix.nagel@cardio.lbg.ac.at (F.N.); bruno.podesser@meduniwien.ac.at (B.K.P.); 7Department of General Surgery, Medical University of Vienna, 1090 Vienna, Austria; andreas.mitterbauer@meduniwien.ac.at (A.M.); michael.bergmann@meduniwien.ac.at (M.B.)

**Keywords:** telocytes, aorta, thoracic ascending aortic aneurysms, exosomes, cellular senescence, miRNA, SMC-phenotype switching

## Abstract

A hallmark of thoracic aortic aneurysms (TAA) is the degenerative remodeling of aortic wall, which leads to progressive aortic dilatation and resulting in an increased risk for aortic dissection or rupture. Telocytes (TCs), a distinct type of interstitial cells described in many tissues and organs, were recently observed in the aortic wall, and studies showed the potential regulation of smooth muscle cell (SMC) homeostasis by TC-released shed vesicles. The purpose of the present work was to study the functions of TCs in medial degeneration of TAA. During aneurysmal formation an increase of aortic TCs was identified in human surgical specimens of TAA-patients, compared to healthy thoracic aortic (HTA)-tissue. We found the presence of epithelial progenitor cells in the adventitial layer, which showed increased infiltration in TAA samples. For functional analysis, HTA- and TAA-telocytes were isolated, characterized, and compared by their protein levels, mRNA- and miRNA-expression profiles. We detected TC and TC-released exosomes near SMCs. TAA-TC-exosomes showed a significant increase of the SMC-related dedifferentiation markers KLF-4-, VEGF-A-, and PDGF-A-protein levels, as well as miRNA-expression levels of miR-146a, miR-221 and miR-222. SMCs treated with TAA-TC-exosomes developed a dedifferentiation-phenotype. In conclusion, the study shows for the first time that TCs are involved in development of TAA and could play a crucial role in SMC phenotype switching by release of extracellular vesicles.

## 1. Introduction

Thoracic ascending aortic aneurysms (TAA) are in most cases asymptomatic, but can present an increased risk of aortic dissection and can consequently lead to death [1]. Anatomy and function of the ascending aorta are complex and dependent on a normal extracellular matrix (ECM). Aortic ECM remodeling can lead to an increase of collagen fibers and loss of vascular smooth muscle cell (vSMC) contractility. The remodeling processes can be induced by chronic oxidative stress [2]. This repetitive cellular stress leads to cellular senescence, which includes the secretion of pro-inflammatory cytokines, growth factors, and extracellular matrix degradation proteins [3]. However, most vSMCs in a healthy aortic wall exhibit a contractile phenotype that maintains vascular tone. During the formation of TAA vSMCs can dedifferentiate into a synthetic phenotype, which is characterized by a decrease in contractile protein expression, degradation of ECM, and increased production of matrix metalloproteinases (MMPs) [4]. This process of vSMC dedifferentiation is called SMC phenotype switching [5]. These characteristic changes of TAA tissue may result in decreased arterial structural stability, thereby increasing the chance for development of aortic aneurysm and leading to a potentially lethal dissection [6,7].

Telocytes (TCs) are a recently defined interstitial cell type [8], and can be found in most organs [8,9]. TCs were detected in a wide range of tissues including the heart and cardiac valves [10,11,12], small blood vessels [13], and in other major organ systems and tissues [14,15,16,17,18,19,20,21,22,23,24,25,26,27]. Most recently, we described the presence of TCs in the human ascending aortic tissue [28]. Moreover, detailed analysis clarified cell marker specificity for CD34, ckit, PDGF-a/b and a-SMA in cell cultures of isolated aortic TCs [28]. With negative staining of CD90 and CD31, they clearly differed from pericytes, and other cells found in aortic tissue [2,28,29]. TCs play different roles from mechanical support to immune surveillance depending on their specific locations within different tissues [19,20]. The morphology of TCs is characterized by spindle- or stellar-shaped small cell bodies, and a variable number of prolongations, called telopodes (Tps). Telopodes, in turn, include thick sections, the podoms, which contain mitochondria, endoplasmic reticulum, and other organelles and the podomers, which are the thin extended sections. Together they form an interstitial network around the vasculature with homo- and heterocellular junctions to release shed vesicles and exosomes, which might have the ability to control the blood vessels. However, the relation of TCs to blood vessels and vSMCs, as well as participation to intercellular signaling, tissue renewal, and regeneration was described previously [22,23,24]. Most widely described expression markers for TCs include CD34, Vimentin, ckit and platelet-derived-growth-factor receptor-^®^ (PDGFR-^®^) [25]. A physiological role for vascular TCs is assumed by their expression of Krüppel-like factor-4 (KLF-4), vascular endothelial growth factor (VEGF), and angiogenic miRNAs [26,28]. TCs express MMP-9 and play an essential role in ECM degradation during angiogenesis [27].

The human aorta is noted to have a highly heterogeneous microenvironment with many CD34 expressing cell types [30], the most important of which are pericytes and endothelial progenitor cells (EPCs). Whereby, pericytes and supra-vasa are double positive for CD34 and CD90 (fibroblast marker), EPCs are positive for CD34 and CD31 [31]. The presence of EPCs in the adventitia is thought to be associated with their potential for neovascularization and capability of smooth muscle lineage progression [29]. In this context, EPCs which are additionally CD133 positive (synonym AC133, Prominin) were described as ‘primitive circulating stem cells’ [32,33]. CD34^+^/CD133^+^ EPCs express high levels of VEGF [34,35], and lose CD133 expression during an early stage of differentiation [36].

Since SMC phenotype switching occurs early in the development of aortic disorders, and the mechanisms by which this occurs are not completely understood, our overall goal was to investigate the regulatory role of TCs on SMC phenotype and aneurysm disorders. We hypothesized that the number of TCs, and their functional role in releasing vesicles plays a crucial role in SMC phenotype regulation during aneurysm formation. Here, we evaluated and characterized TCs and EPCs in patients with either aortic root dilatation or sporadic TAA.

## 2. Results

### 2.1. Increased Number of TCs in Aortic Aneurysm Disease

To investigate the role of TCs in aortic aneurysms, we investigated the number of TCs using immunohistochemistry. Immunofluorescence is an ideal method to measure the occurrence of TCs by detection of telocyte-specific markers in aortic tissue sections, as described previously [28], and TAA samples were obtained from patients undergoing heart transplantation or elective aneurysm surgery. Patient baseline characteristics are presented in Table 1. It should be noted that TAA patients differ with respect to cardiovascular risk factors from patients undergoing heart transplantation. A significance was also observed in chronic renal failure, ejection fraction, and a higher proportion of patients taking aspirin and ^®^-blocker, whereby the significance is due to obtained HTA specimens from heart transplant recipients and their chronic heart insufficiency. Table 2 and Table 3 show patient’s characteristics with focus on aortic diameter and intimal thickness measured by immunohistochemistry.

We identified significantly more CD34^+^/ckit^+^ TCs in tissue from TAA patients (*n* = 29) compared to HTA samples (*n* = 23) (Figure 1A). In detail, we observed an increase of TCs in adventitial, medial, and intimal layers of the diseased aorta (*p* < 0.001) (Figure 1A,B). The highest percentage increase in TC number was detected in the intimal layer of TAA samples (Figure 1A). Moreover, correlation studies show a significance in the number of TCs correlated to the aortic diameter, the higher the aortic diameter of aneurysmal aorta, the higher the presence of TCs (adventitia, R = 0.346 *p* < 0.01; media, R = 0.454 *p* < 0.01; intima, R = 0.2819 *p* < 0.05) (Figure 1C,D). Double staining of well-known TC markers (CD34, ckit, vimentin, and PDGFR-^®^), as well as the lack of endothelial marker CD31 confirmed TC specificity (Appendix A). Notable, the thickness of the tunica intima showed a significant decrease after aneurysmal formation compared to healthy aorta (*p* < 0.001) (Figure 1E). In the non-aneurysmatic aorta of the HTA samples TC were predominantly found in the tunica adventitia as compared to tunica media and tunica intima (*p* < 0.001 and *p* < 0.05, respectively). This corresponds to our previous observations, where aortic TCs were mainly located in the adventitial layer and their perivascular network [28,32]. In summary, during aneurysmal formation and advanced stage of thoracic aortic disease, the number of TCs increased significantly with noticeable distribution of TCs to the intimal layer.

Most recently, CD34^+^/CD133^+^ EPCs and CD34^−^/ckit^+^ hemopoietic stem cells (HSCs) have been described to form a discrete progenitor cell niche for the development of thoracic aortic disease [29]. Immunostaining showed distribution of CD34^+^/CD133^+^ positive cells in aneurysmal tissue (Figure 1F). Further, we observed ckit positive, and CD34 negative cells HSCs in HTA samples, which were significantly decreased after aneurysmal formation in TAA cells (Figure 1G,H).

### 2.2. Comparison, Characterization, and Analysis of Released Exosomes of HTA- and TAA-TCs

To further investigate various protein expression markers in HTA and TAA samples, TCs were isolated from HTA and TAA samples as previously described [8,9,28] (Figure 2A). Cells were sorted based on CD34 and ckit protein expression and purification of CD90 negative cells to distinguish TCs from dedifferentiated vSMCs or fibroblasts. After reaching ~80% of confluence at cell culture, we performed mRNA and protein analysis. The TC phenotype was characterized by mRNA expression of ckit, vimentin, PDGFR-α/-^®^, KLF-4, and CD29/integrin β–1 in TCs (Figure 2B). TCs isolated from TAA specimens showed a significant increase of vimentin (*p* < 0.01), PDGFR-α (*p* < 0.05), and KLF-4 (*p* < 0.01) compared to TCs isolated from HTA specimens. Moreover, Western blot analysis was conducted in two individual HTA-TC cell cultures and two individual TAA-TC cell cultures, to analyze protein levels of TC expression markers as well as vSMC-dedifferentiation markers (Figure 2C). This revealed higher protein levels of ‘contractile’ SMC-phenotype marker SM-calponin, α-SMA in HTA-TCs, whereby ‘synthetic’-marker vimentin and KLF-4 were decreased in HTA-TCs compared to TAA-TCs. Using TEM, we observed a close interaction of TCs via their telopodes with endothelial cells (ECs), fibroblasts (FBs), and vSMCs. Interestingly, we found exosome-containing multivesicular carriers (MC) primarily located in TC podomeres (Figure 2D(a)), suggesting a paracrine activity of TCs. Higher magnifications showed possible communication between TC and vSMC with synthetic phenotype morphology (Figure 2F(a–c)). We observed an invagination of telopodes by vSMC and formed caveolae close to the cell convergence of TC and vSMC.

We next isolated exosomes from HTA and TAA cell cultures and analyzed their phenotype using immunoblotting (Figure 2G–I). The quality, concentration (particle per ml) and particle diameter (nm) of exosomes was confirmed by qNano analysis (Figure 2G). Exosomes were characterized by KLF-4 and VEGF-A protein. These proteins were increased in TAA cell cultures versus HTA cultures (Figure 2H and Appendix A). Exosomal surface marker CD34, CD63, HSP90, and TSG101 revealed exosome purity (Figure 2H,I). MicroRNA (miR) expression profiles were performed to analyze miRs involved in phenotype-switching of vSMCs (Figure 2F). TCs isolated from TAA showed a significant increase of miR-146a, miR-221, and miR-222, which were previously described for dedifferentiated vSMC, *p* < 0.001, *p* < 0.01, and *p* < 0.01, respectively (Figure 2J). Whereas miR levels found in contractile vSMCs were decreased, miR-143 and miR-145, *p* < 0.05 and *p* < 0.05, respectively. In TAA-exosomes, reduced levels of miR-21 (*p* < 0.05) and miR-145 (*p* < 0.01), and increased levels of miR-146a (*p* < 0.01), miR-221 (*p* < 0.05), and miR-222 (*p* < 0.001) confirmed miR levels which are mainly found in dedifferentiated vSMCs.

### 2.3. Exosomes Isolated from TCs Influence vSMC Phenotype Characteristics

We then investigated whether exosomes, isolated from HTA- versus TAA-cultured TCs, had a regulatory potential to markers involved in SMC dedifferentiation. When vSMC were cultured in the presence of exosomes isolated from TAA-TCs they tended to become less spindle-shaped, and to develop the more irregular morphology associated with synthetic vSMCs, compared to exosomes isolated from HTA-TCs or exosome isolation procedure from cultured vSMCs as control (Figure 3A). vSMCs treated with TAA-exosomes showed a trend of decreased mRNA expression of smooth muscle-cell myosin-heavy-chain 11 (*SMMHC*), *-SMA*, and *SM-calponin* compared to control groups (Figure 3C). TAA-exosome treatment of vSMCs significantly increased mRNA expression of SMC-dedifferentiation markers *collagen-1* (*p* < 0.01), *vimentin* (*p* < 0.01), and *KLF-4* (*p* < 0.01) (Figure 3B). Cell-metabolism assays (MTT) and cell proliferation assays demonstrated an increase of vSMC cell proliferation in vSMCs treated with TAA-exosomes (*p* < 0.01), compared to control group (Figure 3C,D). MTT assays were conducted 1 and 4 days after treatment start (Figure 3C). Moreover, these effects were very similar to those observed in samples which dedifferentiation process of vSMCs were induced by recombinant PDGF-BB protein in a concentration of 20 ng treatment [36]. Cell proliferation assay was used to exclude off-target effects by several dilutions of undiluted (0), 2-times, 4-times, and 6-times exosome concentrations compared to control supernatant of vSMCs (Figure 3D).

Similar to RT-qPCR results, ELISA measurement of *collagen-I* confirmed an increase of protein concentrations after TAA-exosome or PDGF-BB treatment, *p* < 0.01 and *p* < 0.001, respectively (Figure 3E). Additionally, wound healing assay was performed on vSMCs treated with isolated TC-exosomes or TC-conditioned medium (TCM) (Figure 3F,G). Migration distance of vSMCs was increased after 24 h of treatment with TAA-exosomes (*p* < 0.01), whereby, a significant increase was observed after treatment with both ‘whole’ cell culture supernatant of sorted HTA- and TAA-TCs, compared to specific controls (*p* < 0.01) (Figure 3F). The assumption that exosomes may initiate or regulate SMC-dedifferentiation would merit gain-of-function and loss-of-function analysis of miR.

## 3. Discussion

The adventitia of the ascending thoracic aorta represents a specialized perivascular niche. The occurrence of TCs in aortic human tissue has been demonstrated previously [28]. For the first part, we classify that TCs are playing a crucial role in enhancement of aneurysm formation depending on the expression profile of aortic TCs and their released exosomes. For the first time, we classify TCs by their antigenic profile, function, and location associated with aneurysm formation and show a significant increase of TCs in the diseased aorta, which correlated to advanced pathogenicity. Based on the expression profile of TCs and the high occurrence of well-known factors during aneurysmal formation (e.g., KLF-4 and VEGF-A) [1], we characterize their potential for smooth muscle lineage progression. This finding is supported by a cell culture experimental subset showing that the treatment of vSMC with TC-exosomes leads to dedifferential-phenotype changes. This release of TC-related factors is involved in vSMC phenotype switching, which could play a crucial role in the development of instable aortic tissue.

Popescu et al. discussed a TCs stromal progenitor cell analogy, which means that TCs can participate in immune surveillance and mesenchymal differentiation functions [37]. However, the classification and functional characterization of vasa-vasorum-associated perivascular progenitor cells in human aorta describe a subset of CD34^+^/CD31^+^/α-SMA^−^ endothelial progenitor cells which are mainly abundant in aortic adventitia [29]. The functionality of EPCs is described in neovascularization of cardiovascular diseases [36,38]. Our current finding of the regulatory function of TCs and their markers’ similarity to HSCs such as EPCs, provide additional support that these unique cell populations may play a distinct and important role in aortic diseases [39]. Concordantly, we found that the onset of CD34^−^/ckit^+^-progenitor cells were decreased with disease progression, and CD133^+^/CD34^+^ double positive cells were detected in aortic media. However, the molecular identification of TCs and the distinction from endothelial progenitor cells (EPCs) are presented by negative staining of CD34^+^/ckit^+^/CD133^−^ for TCs. Nevertheless, it remains open how an attraction of TCs from the adventitial layer to the intimal layer occurs. Is the attraction of EPCs in aneurysmatic tissue, the origin for TCs, or are they only involved in the induction of TCs? Further studies are required to clarify the relationships between the functionality of EPCs or the infiltration and differentiation of EPC-subsets to TCs, regarding the near identically expression profile.

In recent years, the therapeutic effect of exosomes derived from TCs have been investigated intensely in multiple disease models and show that these exosomes exert functions similar to those of stem cells, including promoting tissue remodeling and expression of pro-angiogenic miRNA that regulate tissue repair via a paracrine-mediated mechanism in the vasculature [34]. However, to date, few studies have aimed to determine the functional role of exosomes derived from TCs in angiogenesis and tissue remodeling in vascular disorders [9,34]. In this study, we frequently observed exosomes in the immediate vicinity of pits, which suggests that the endocytosis of these vesicles may pass messages from one cell to another through exosomes. Thus, we isolated exosomes from TCs and evaluated their functions. Since it has been reported that TCs have proangiogenic functions, we hypothesized that exosomes of TCs would exert an influence on SMCs. The diversity of vSMC function is reflected in their contractile and synthetic phenotype, which are characterized by substantial differences in marker expression, morphology, and activity [34,40,41,42]. When TCs derived from aneurysmal human tissue were compared to those of healthy aortic tissue, we detected an increase of specific mRNA expression for a synthetic vSMC phenotype. Whereby markers for a contractile vSMC phenotype were downregulated in western blot. HTA-exosomes and TAA-exosomes are equally found in the *vimentin* and *KLF-4* mRNA expression, their cell metabolism and proliferation, whereas TAA-exosomes interestingly show more disrupting features of diseased aortic tissue, collagen secretion, and regenerative potential (Figure 3), which correlates with previous findings of aortic cells found during aneurysm formation. The development of a TAA or HTA is a process of several pathomechanisms, still not clearly understood. Initial triggers release a destructive process of oxidative stress, apoptosis or dedifferentiation of vSMCs, and proteolytic fragmentation of the ECM. The same triggers are found to release TCs exosomes. The adverse environment now increases the reactivity and boosts oxidative stress by producing reactive nitrogen and oxygen species, which aggravates apoptosis or dedifferentiation of vSMCs leading to aneurysmal formation. Our findings, supported by a cell culture experimental subset, show that the treatment of vSMC with TC-exosomes leads to dedifferential-phenotype changes. Further investigations are needed to identify a direct link between oxidative stress and TCs exosome release leading to aneurysmal formation.

The characterization and comparison of exosomes derived from HTA- and TAA-TCs revealed a high amount of VEGF-A and KLF-4 proteins in shed vesicles. Further, the human genome encodes 1048 miRNAs, which virtually regulate all biological processes [43]. Specific miRNA expression patterns have been previously described for TC-exosome treatment, where they were responsible for complex regulatory function driven by telocytes [44]. In our study, miRNA in aortic TC and TC-released exosomes showed an expression profile which is consistent to previous observations [38,45], but in TAA-samples we detected an increased shift of miRNAs involved in dedifferentiation of vSMCs (*miR-146a*, *miR-221* and *miR-222*) [46]. Keeping in mind some of the roles attributed to the TC, such as the juxta/paracrine activity, the ability to remodel the collagen fibrils and to control tissue homeostasis [47], it could be speculated that the increased expression of vimentin, PDGFR-a and KLF-4 represents a potentiation of these functions. Specifically, KLF-4 is upregulated by shear stress [29,41], a typical EC differentiation stimulus found during aneurysm formation, and inhibits SMC maturation [48]. When we cultured vSMCs in the presence of exosomes isolated from TCs, we observed a dedifferentiation phenotype, which includes cell morphological changes, increased metabolism, and a significant increase of synthetic-phenotype related mRNA expression in vSMCs (*KLF-4*, *vimentin*, and *collagen-I*). The synthetic phenotype of vSMCs plays a crucial role in progressive aneurysm formation in human and is associated with high expression of VEGF-A, vimentin, KLF-4, and ECM degrading enzymes [37]. Besides our findings and others of vimentin and VEGF-A expression in TCs, most recently the expression of metalloproteinase-9 (MMP-9) was also attributed to TC [28,29]. MMP-9 is essential for degradation of ECM components [39].

In conclusion, the study shows for the first time that TCs are involved in development of TAA. Whereby the significantly high number of TCs found in TAA seems to be the decisive factor leading to an imbalance of homeostasis and to an uncontrolled remodeling of the tissue. The characterization of their exosome-related function and location in TAA, as well the observation of progenitor cell recruitment of EPCs-subsets, show the potential of aortic TCs for involvement in smooth muscle lineage progression during aneurysm formation. Our results provide preliminary evidence that aortic TCs have therapeutic potential for the treatment of TAA and the prevention of fatal progression of the disease.

## 4. Materials and Methods

### 4.1. Patient’s Specimens

Human aortic tissue samples (52 samples) were obtained either during heart transplantation (23 samples), or during surgical procedure which involved aneurysm surgery of TAA (29 samples). Patients with ongoing endocarditis, sepsis, recent infectious disease, or genetic disorders (e.g., Marfan’s syndrome) were excluded. Additionally, the intake of immunomodulation therapy (e.g., cortisone) or anti-tumor therapy was an exclusion criterium. After receiving the specimens, aortic tissue was sliced, and one part was snap frozen and stored in liquid nitrogen, one part was fixed in 4.5% formalin, one part in 2.5% glutaraldehyde. The remaining tissue was subjected to cell isolation. This study was approved by the Ethical Committee of the Medical University of Vienna (EK 1280/2015). Written informed consent was obtained from all patients prior to inclusion in the study. The investigation conformed to the principles that are outlined in the Declaration of Helsinki regarding the use of human tissue.

### 4.2. Isolation and Sorting of Aortic Telocytes, Fibroblasts and vSMCs

Isolation of human fibroblasts (*n* = 3), vSMCs (*n* = 4), and TCs (HTA, *n* = 10; TAA, *n* = 10) was performed according to our established protocol with few modifications as outlined in this section [28]. Briefly for isolation of TCs, aortic tissue was collected in RPMI-1640 cell culture medium supplemented with 10% fetal bovine serum [FBS], 25 mM HEPES as well as 100 IU/mL penicillin, and 100 UI/mL streptomycin (medium and all supplements were obtained from Gibco/Life Technologies Ltd., Pailey, UK). Aortic samples were dissected and minced into small pieces of about 1 mm^3^ and incubated for 3 h at 37 °C with mixture of collagenase type IV (Gibco) and elastase (porcine pancreas, Calbiochem/Merk, Darmstadt, Germany) dissolved in TC-cell culture medium (TC-CCM): high glucose (HG)-DMEM (Lonza Bioscience Solutions, Cologne, Germany) supplemented with 1.5 mM HEPES and 20% FBS (Gibco/Life Technologies Ltd., Vienna, Austria). The isolated cells were filtered through a cell strainer (100 µm), centrifuged and re-suspended in TC-CCM. 90 min after seeding, the supernatant, which mainly contains the majority of TCs, was removed and transferred into a new 24-well plate containing TC-CMM. Cells were cultivated at 37 °C in humidified atmosphere (5% CO_2_). The morphology of TCs was observed and pictures taken using a phase-contrast microscope (Olympus CKX41 with Olympus SC-20 camera, Olympus Life Science, Vienna, Austria).

For CD34^+^/ckit^+^/CD90^−^ TC-cell sorting to distinguish TCs from dedifferentiated vSMCs or fibroblasts., cultured aortic cells were collected in FACS buffer (PBS including 0.1% FBS), and 25 mM HEPES was added to the FACS buffer to prevent it from becoming basic and maintain the pH between 7.0–8.0, and 1 mM–5 mM EDTA to the buffer to prevent formation of aggregates. Cells were stained with 1× or 0.5× of the antibody concentration used for immunocytochemistry, followed by appropriate secondary antibody, if necessary (Appendix A). Cells were re-suspended at a concentration of 2–3 × 10^7^ cell/mL. Immediately before sorting, cells were filtered through a 70 µm mesh filter to prevent clogging and collected in HG-DMEM supplemented with 30% FBS afterwards. Cells were analyzed directly by western blot or cultivated in standard culture medium depending on the cell type (see above). Cell sorting was performed with the BD FACSAria™III Fusion (Software: BD FACSDiva Version 8.0.2, Becton, Dickinson and Company, San Jose, CA, USA).

### 4.3. Immunofluorescence Staining and Microscopy

For immunocytochemical staining, cells were grown on 8 chamber slides (Falcon^®^ glass slide with polystyrene vessel, Fa. Falcon/Szabo Scandic, Vienna, Austria) and fixed in 4% paraformaldehyde for 10 min. Followed by permeabilization in 0.1% saponine, and blocked with PBS (ThermoFisher Scientific, Waltham, MA, USA) containing 1% bovine serum albumin (BSA), 10% goat serum and 0.3 M Glycine for 1 h at 37 °C. Samples were incubated with 2–5 µg of primary antibody O/N according to the listed working dilutions (Appendix A), followed by incubation with an appropriate secondary antibody including 1 µg/mL Dapi (ThermoFisher Scientific, Waltham, MA, USA) or 4 µg/mL Hoechst 34580 (Bio-Connect B.V., TE Huissen, The Netherlands), and mounted in Prolong Gold Antifade (Molecular Probes, ThermoFisher Scientific, Waltham, MA, USA). Negative controls were obtained following the same protocol, but omitting the primary antibodies, and the usage of purified anti-mouse and anti-rabbit IgG (Abcam, Cambridge, UK).

For immunohistological staining, aortic tissue samples were fixed in 4% PBS-buffered formaldehyde. The tissues were embedded in paraffin, deparaffinized with HistoSAV and rehydrated in a descending series of ethanol. Following heat-induced antigen retrieval with citrate-buffer (pH 6), the sections were blocked (10% goat serum, 1% BSA, and 0.1% Tween-20 in PBS) at RT for 60 min. The antibody incubations corresponded to ICC staining protocol (see above). The density of TCs was calculated as the mean of total number of TCs/total number of DAPI stained nuclei per cross section.

For confocal microscopy, we used a LSM700 Meta microscopy laser system, the appropriate filters, and a ZEN 2010 microscopy system (Zeiss, Inc. Jena, Munich, Germany). For spot counting and co-localization analysis images were analyzed with the CellProfiler™ cell image analysis software.

### 4.4. Transmission Electron Microscopy

Samples of the aortic wall of approx. 2 cm^2^ were fixed immediately after surgery in 2.5% glutaraldehyde. After 6 h, samples were cut into smaller pieces of 1 mm^3^ and washed three times in 0.1 M cacodylate buffer. The secondary fixation was carried out either for 2 h. in 2% osmium tetroxide/0.1 M cacodylate buffer or for 2 h. in 1% reduced osmimum tretroxide, both at room temperature. Dehydration and embedding in Epon resin followed standard procedures. Ultrathin sections (70 nm) were cut with a Reichert UltraS microtome and contrasted with uranyl acetate and lead citrate. Images were acquired with a FEI Tecnai20 electron microscope equipped with a 4 K Eagle CCD camera and processed using the Adobe Photoshop software package.

### 4.5. Microvesicle and Exosome Isolation

Microvesicle and exosome isolation was performed as previously described [28]. Briefly, HTA-TC-, TAA-TC-, and vSMC-cells were grown in FCS-free culture medium for 24 h. The cell suspension was centrifuged at 480× *g* at 4 °C for 5 min to remove any intact cells, followed by a 3200× *g* spin at 4 °C for 20 min to remove dead cells. To isolate shedding microvesicles (sMVs), the supernatant was centrifuged at 10,800× *g* at 4 °C for 20 min in an Optima L80 ultracentrifuge with a SW41Ti rotor (Beckman Coulter, Mississauga, ON, Canada). The pellet, containing sMV, was washed once with PBS^−/−^ and ultracentrifuged at 10,800× *g* for 30 min. The pellet was dissolved in fresh medium for immediate use or stored at −80 °C for western blot analysis. The remaining culture medium was transferred to ultracentrifuge tubes and sedimented at 110,000× *g* at 4 °C for at least 75 min. The supernatant constituting exosome-free medium was removed and the pellets containing exosomes plus proteins from media were resuspended in PBS. The suspension was centrifuged at 100,000× *g* at 4 °C for at least 60 min to collect final exosome pellets. The quality of exosomes was confirmed by qNano analysis (Izon Science Ltd., Oxford, UK). Protein content of the exosome pellet was quantified using the Bradford protein assay kit (Biorad, Hercules, CA, USA). Pellet was dissolved in vSMC-specific cell culture medium for cell growth analysis and scratch assay, or the pellet was analyzed for miRNA and protein detection.

### 4.6. ELISA, Wound Healing Assay and EZ4U Measurements

Collagen-I level was measured with a Soluble Collagen Assay Kit according to manufacturer instructions (ab241015, Abcam, Cambridge, UK). For wound healing experiments a scratch assay was used. Indicated cells were plated with ~80% intensity in 6-wells and after attachment, medium was changed after 24 h. Scratch was conducted in an appropriate size and cells were washed and treated with TC- or vSMC conditioned culture medium (CCM) from separate cell cultures, or with isolated exosomes resolved in appropriate cell culture medium. Images were conducted after 0, 12, and 24 h. Cell viability and cell proliferation was assessed using an EZ4U kit (Biomedica MP, Vienna, Austria) or cell counting kit (CCK)-8 assay (Sigma-Aldrich, Taufkirchen, Germany) according to manufacturer’s instructions. Cells were seeded in ~25% cell density and after attachment, cells were treated with indicated exosomes or vSMCs-control dissolved in vSMC culture medium. For EZU4 assay, the measurements were conducted 24 h. and 96 h. after treatment start. For CCK-8 assay, measurements were conducted 72 h. after treatment start. For cell growth analysis and collagen-I measurements, a recombinant human 20 ng PDGF-BB protein (ab79746, Abcam, Cambridge, UK) was used as positive control.

### 4.7. miRNA and mRNA Isolation and Real-Time PCR (RT-qPCR)

For mRNA of all samples, RNA was isolated using Trizol (PeqGOLD TriFast, Peqlab, VWR, Vienna, Austria) followed by purification with the E.Z.N.A. Microelute Total RNA Kit (Omega Bio-Tek, VWR, Vienna, Austria), including the optional DNA digestion step (RNase-free DNase I Set, Omega Bio-Tek, VWR, Vienna, Austria) according to manufactures’ instructions. For RT-qPCR, RNA was reverse transcribed using the QuantiTect Reverse Transcription Kit (Qiagen, Hilden, Germany), followed by qPCR with the GoTaq RT-qPCR Master Mix (Promega, Mannheim, Germany) according to manufacturer’s instructions.

For miRNA measurements, RNA was isolated from exosome pellet with miRNeasy kit (Quiagen, Hilden, Germany) according to manufacturer’s instruction. cDNA was generated with the miScript II RT Kit and was used as a template for real-time PCR with the miScript SYBR Green PCR Kit (Quiagen, Hilden, Germany) in accordance with the manufacturer’s protocol and a gene-specific probe in a 7500 Real-Time PCR system (Applied Biosystems, Foster City, CA, USA). The relative expression level for each miRNA was computed using the comparative CT method [31]. miRNA expression was normalized to small nucleolar RNA U6. For mRNA analysis, samples were normalized to the geometric mean of two reference genes (*GAPDH*, *RPLP0*). Primer sequences are listed in Appendix A.

## Figures and Tables

**Figure 1 ijms-23-04730-f001:**
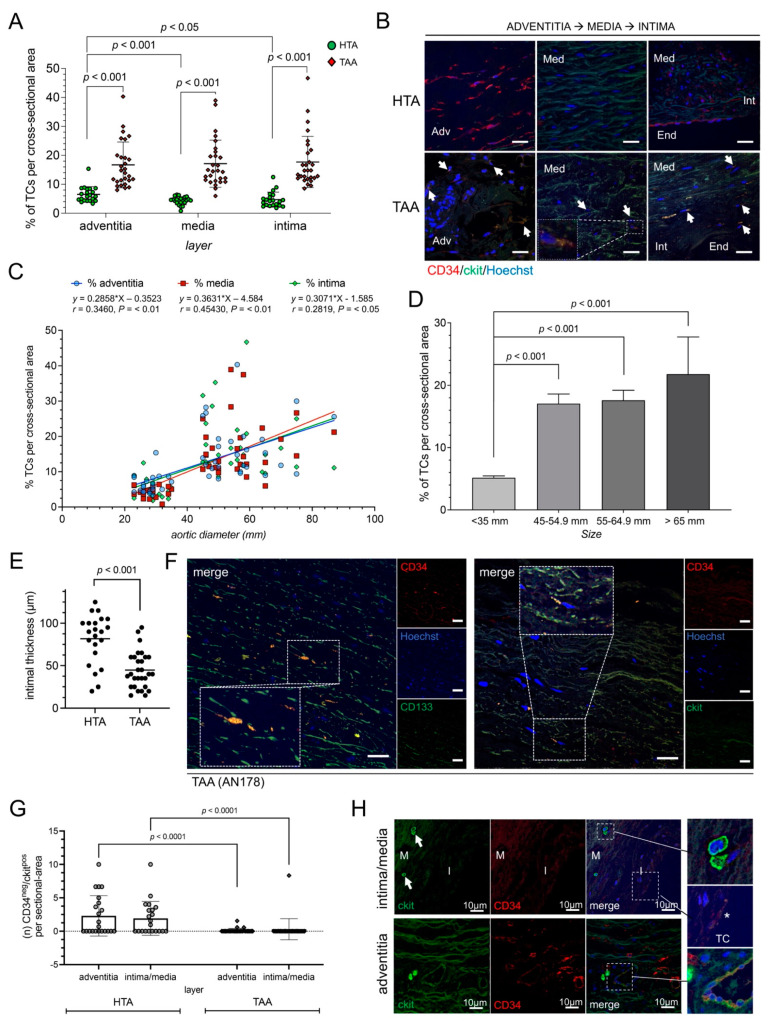
Human aneurysmal tissue showed an increase of TCs and EPCs depending on aneurysm size. (**A**) Statistical analysis of TCs per cross-sectional (number of CD34^+^/ckit^+^ TCs/number of total cells) in healthy thoracic aortic (HTA, *n* = 23) or thoracic ascending aneurysm (TAA, *n* = 29) tissue. Specific layers of aorta are shown. (**B**) Representative image of double-positively TCs in all three aortic layers, adventitia (Adv; on the left), media (Med; in the middle), and transition zone of media, intima (Int), and endothelium (End; on the right), are presented. TCs are indicated by narrows. Magnification shows morphology of a TC in medial layer. Scale bar, 20 µm. CD34, red; ckit, green; cell nuclei, Hoechst, blue. (**C**) Correlation study through Pearson’s linear regression analysis of the aortic diameter in mm and percentage of TCs per cross sectional area. Location of TCs was separated into adventitial (adv, blue), medial (med, red) and intimal (int, green) layer. R and *p* values are given on the top. (**D**) Diagram of TCs detected depending on aortic diameter (size in mm) in healthy thoracic aortic (HTA, *n* = 23) or thoracic ascending aneurysm (TAA, *n* = 29) tissue. Data are mean ± SD. (**E**) Intimal thickness was reduced in TAA samples (*n* = 29) compared to HTA (*n* = 23). Intimal thickness is given in µm. (**F**) Representative images of immunostaining of EPCs (left) located in medial layer were double stained with CD133 (green) and CD34 (red) with TC-like morphology. CD34^+^/ckit^+^ TCs staining (CD34, red; ckit, green) is shown on the right. Single immunofluorescence images are presented on the right of each merged image. Nuclei were counterstained with Hoechst (blue). Scale bar, 50 µm. Statistical analysis in healthy thoracic aortic (HTA, *n* = 23) or thoracic ascending aneurysm (TAA, *n* = 29) tissue (**G**) and representative image (**H**) of ckit-positive and CD34-negative HSC-subset detected in human aortic tissue. CD34^−^ EPCs were analyzed and given for intima/media and adventitia of aortic vessel separately (**G**). (**H**) Immunofluorescence showed ckit^+^ EPCs next to endothelial tube with double-positive TCs. Magnifications present morphological round to oval ckit^+^ EPC (right upon), double-positive TC (*) (right middle), and EPC close to endothelial tube with TCs (right bottom). CD34, red; ckit, green; nuclei, blue (Hoechst).

**Figure 2 ijms-23-04730-f002:**
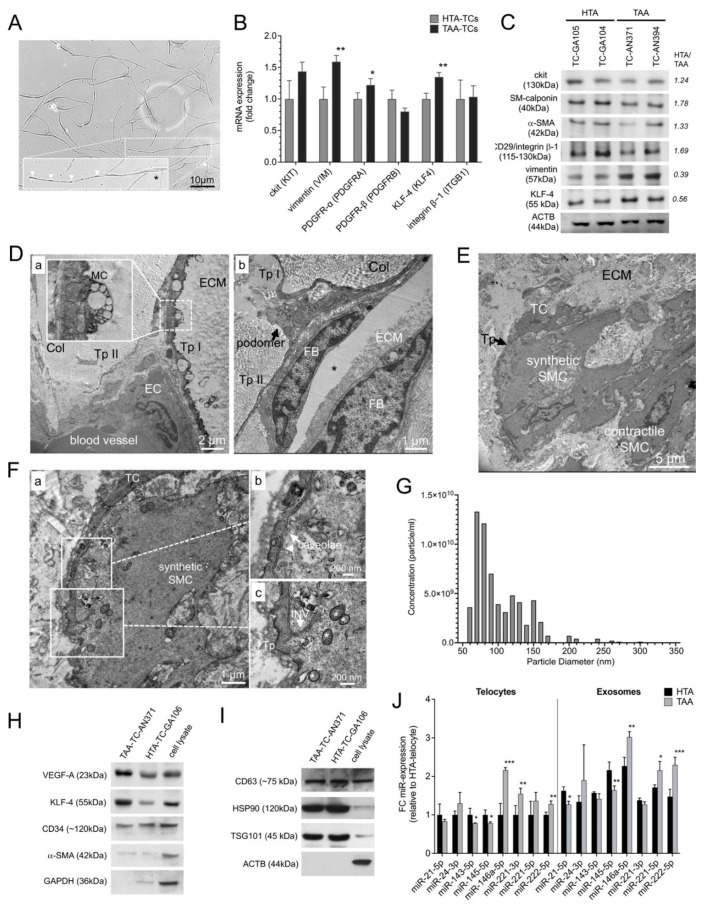
Characterization of isolated and sorted TCs and exosomes from HTA (*n* = 23) and TAA (*n* = 29) specimens. (**A**) Representative image of isolated and CD34^+^/ckit^+^ sorted TCs from aortic tissue. In the magnification TC showed typical morphology with oval cell body (Asterisc), long thin processes including intermitted telopodes (arrows) shown by light microscopy 14 days after isolation. (**B**) mRNA expression profile of aortic TC marker genes (*KIT*, *VIM*, *PDGFRA*, *PDGFRB*, *KLF4*, and *ITGB1*) confirmed TC phenotype and showed differences between cells isolated from healthy aortic specimens (HTA-TCs, *n* = 10) compared to aneurysmal aortic specimens (TAA-TCs, *n* = 10). Bars indicate the relative expression of each mRNA normalized to *GAPDH* and *RPLP0*. Data are mean ± SD of three independent experiments. (**C**) Western blot analysis of TCs isolated from two individual HTA-aortic samples and two from TAA-aortic samples. Six protein markers which are involved in phenotype switching of SMCs were analyzed (ckit, SM-calponin, α-SMA, CD29/integrin ^®^-1, and KLF-4). ACTB was used as loading control; Primary antibodies and the observed molecular weight (kDa) are given on the left. Statistical calculation given by HTA/TAA ratio are given on the right. (**D**–**F**) Representative transmission electron micrographs of medial layer in human aortic specimens. The connections between TCs and (**D**-**a**) endothelial cell (EC), and (**D**-**b**) fibroblast (FB) embedded between collagen fibers (Col) is shown. Preparation artifact (Asterix). (**E**) Cell convergence of TC and vSMC from synthetic phenotype is shown. vSMC-phenotype was characterized by their found intracellular filament order. (**F**) Magnifications of TC telopode (Tp) to synthetic vSMC (**a**) revealed active intake of caveolae (**b**) and invaginated telopode (INV) and mitochondria (asterisks) by vSMC (**c**). MC, multivesicular cargos; ECM, extracellular matrix; TC, telocyte. (**G**) The quality, concentration (particle per ml) and particle diameter (nm) of exosomes was confirmed by qNano analysis (Izon instrument, UK) (*n* = 7). (**H**,**I**) Representative western blots of isolated HTA- and TAA-exosomes are shown. Cell lysate was used as control. Exosome-specific soluble factors (VEGF-A, KLF-4, CD34 and α-SMA) (**H**) as well as surface proteins (CD63, HSP90 and TSG101) (**I**) were analyzed. ACTB and GAPDH were used as loading control; Primary antibodies and the observed molecular weight (kDa) are given on the left of each blot. (**J**) Micro RNA (miR) expression profile in HTA (black, *n* = 23) and TAA (grey, *n* = 29) isolated exosomes. Expression of some miR which are involved in SMC-phenotype switching were measured by qRT-PCR. Bars indicate the relative expression of each miR normalized to U6 small nuclear RNA (*RNU6B*) and *SNORD44*. Data are mean ± SD of two independent experiments. * *p* < 0.05; ** *p* < 0.01; *** *p* < 0.001.

**Figure 3 ijms-23-04730-f003:**
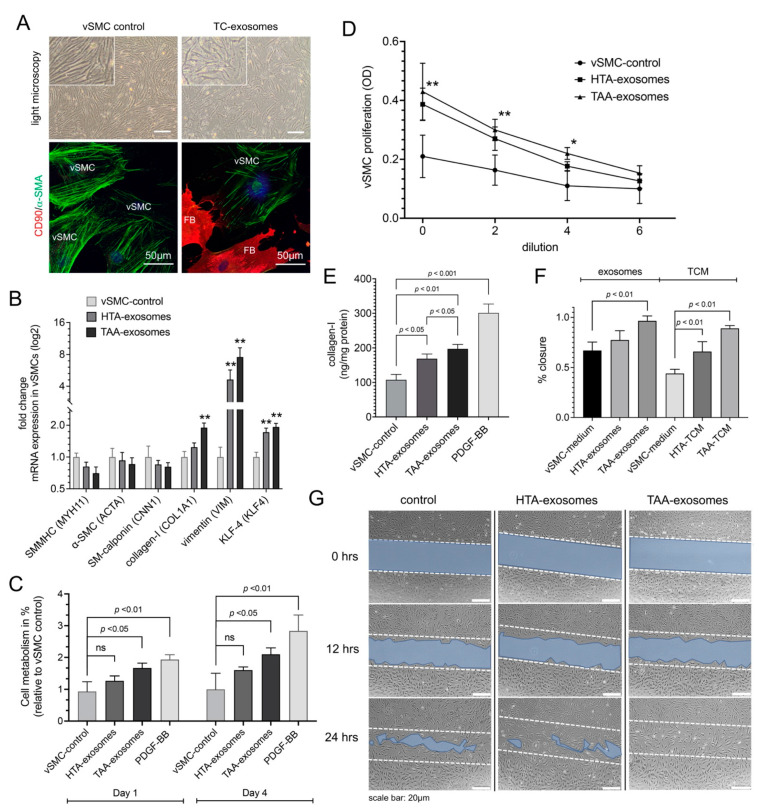
Effect of exosomes isolated from aneurysmal telocytes (TCs) on proliferation, metabolism, and phenotype specific mRNA expression of aortic vascular smooth muscle cell (vSMC)-cell culture. Exosomes were isolated from isolated TCs from healthy aortic tissue (HTA, *n* = 10) or thoracic ascending aneurysm (TAA, *n* = 10). (**A**) Morphological changes shown by light microscopy and α-SMA reengagement by immunofluorescence in SMCs treated with TC-exosomes compared to SMC control in isolated aortic cell culture containing vSMCs and fibroblasts (FB). CD90 (FB-marker), red; α-SMA (vSMC-marker), green; DAPI (cell nucleus), blue. Scale bar of light microscopy 20 µm. (**B**) Quantitative RT-PCR of SMC-phenotype specific mRNA expression after HTA-exosomes or TAA-exosomes compared to vSMC-control. Synthetic phenotype mRNA expression (*COL1A1*, *VIM*, and *KLF4*) were increased, whereby contractile phenotype mRNA expression (*MYH11*, *ACTA*, and *CNN1*) showed no differences after treatment with TAA-exosomes. Bars indicate the relative expression of each mRNA normalized to *GAPDH* and *RPLP0*. Data are mean ± SD of two independent experiments. **, *p* < 0.01. (**C**) Effect of HTA- and TAA-exosomes compared to vSMC-control tested in aortic vSMCs by MTT assay after 1 and 4 days of treatment. PDGF-BB treatment was used as positive control. ns, non-significant. Data are mean ± SD of four independent experiments. (**D**) Dilution-depended exosome-induced cell proliferation tested in aortic SMCs. OD values resulting from CCK-8 assay, HTA- and TAA-exosomes were compared to vSMC control. OD, optical density. Data are mean ± SD of four independent experiments. *, *p* < 0.05; **, *p* < 0.01. (**E**) Collagen-I measurements after different exosome or control treatments after 3 days tested in vSMCs with PDGF-BB as positive control. Data are mean ± SD of three independent experiments. (**F**,**G**) Cell migration (scratch wound healing assay). (**F**) Values of percentage wound closure ± SEM (*n* = 3). Different exosome treatment or TC-conditioned medium (TCM) treatment were compared to vSMC medium as control group 24 h after treatment start. (**G**) Representative images are shown from three independent experiments at time points beginning (0 h), 12 h, or 24 h. Blue area defines the areas lacking cells, initial scratch line shown by dashed lines (wound area, ImageJ). Scale bar, 20 µm.

**Table 1 ijms-23-04730-t001:** Characteristics of the study population.

	Study Population		HTA		TAA		*p* Value
	(*n* = 52)		(*n* = 23)		(*n* = 29)		
Demographic, risk factors, and comorbidities							
Age (years) (range)	58.6	(20–79)	52.2	(20–69)	63.8	(36–79)	**<0.01**
female, *n* (%)	15	(28.8)	5	(18.9)	10	(34.5)	0.26
Body mass index (BMI), *n* (range)	26.8	(18–41)	24.9	(19–30)	28.3	(18–41)	**<0.01**
Adipositas (BMI > 30), *n* (%)	11	(21.2)	2	(8.7)	9	(31.0)	**<0.05**
Smoker, *n* (%)	10	(19.2)	0	(0)	10	(34.5)	**<0.01**
Hypertension, *n* (%)	32	(61.5)	9	(39.1)	23	(79.3)	**<0.01**
Dyslipidaemia, *n* (%)	24	(46.2)	10	(43.5)	14	(48.3)	0.42
Chronic renal failure, *n* (%)	10	(19.2)	8	(34.8)	2	(6.9)	**<0.01**
Diabetes, *n* (%)	6	(11.5)	3	(13.0)	3	(10.35)	0.39
COPD, *n* (%)	9	(17.3)	2	(8.7)	7	(24.1)	0.07
Positive family history, *n* (%)	2	(3.8)	1	(4.4)	1	(3.5)	0.44
Ejection fraction (<50%), *n* (%)	30	(57.7)	23	(100)	7	(24.1)	**<0.01**
Therapeutics							
Oral diabetes therapy, *n* (%)	3	(5.8)	2	(8.7)	1	(3.5)	0.23
Statins, *n* (%)	15	(28.8)	9	(39.1)	6	(20.7)	0.13
Aspirin, *n* (%)	16	(30.8)	10	(43.5)	6	(20.7)	**<0.05**
Beta-Blocker, *n* (%)	25	(48.1)	16	(69.6)	9	(31.0)	**<0.01**
ACE-Inhibitor, *n* (%)	20	(38.4)	9	(39.1)	11	(37.9)	0.47

COPD, chronic obstructive pulmonary disease.

**Table 2 ijms-23-04730-t002:** Correlation coefficients (*r*) of localization of TCs to patient’s baseline characteristics.

	% Telocytes in T. Adventitia		% Telocytes in T. Media		% Telocytes in T. Intima	
	Correlation Coefficients (*r*)	*p* Value	Correlation Coefficients (*r*)	*p* Value	Correlation Coefficients (*r*)	*p* Value
Age	0.369	**<0.01**	0.283	**<0.05**	0.371	**<0.01**
Gender	0.317	**<0.05**	−0.026	n.s.	0.177	n.s.
Body mass index (BMI)	0.344	**<0.01**	0.342	**<0.01**	0.134	n.s.
Adipositas (BMI >30)	0.348	**<0.05**	0.316	**<0.05**	0.046	n.s.
Smoker	0.416	**<0.01**	0.273	n.s.	−0.113	n.s.
Hypertension	0.373	**<0.01**	0.431	**<0.01**	0.334	**<0.05**
Dyslipidaemia	0.175	n.s.	0.121	n.s.	0.051	n.s.
Statins	−0.075	n.s.	0.092	n.s.	−0.051	n.s.
Chronic renal failure	−0.152	n.s.	−0.240	n.s.	−0.240	n.s.
Diabetes	0.072	n.s.	0.172	n.s.	0.079	n.s.
Oral diabetes therapy	0.014	n.s.	0.006	n.s.	0.042	n.s.
COPD	0.122	n.s.	0.333	**<0.05**	0.190	n.s.
CVD	−0.070	n.s.	0.136	n.s.	−0.071	n.s.
Ejection fraction (<50%)	−0.420	**<0.01**	−0.439	**<0.01**	−0.360	**<0.05**
Aspirin	−0.188	n.s.	−0.260	n.s.	−0.136	n.s.

COPD, chronic obstructive pulmonary disease; CVD, coronary vessel disease; n.s., non-significant.

**Table 3 ijms-23-04730-t003:** Aneurysm size characteristics of the study population.

	Study Population		HTA		TAA		*p* Value
	(*n* = 52)		(*n* = 23)		(*n* = 29)		
Thoracic aorta ascendens:							
Size < 35 mm, *n* (%)	23	(44.2)	23	(100)	0	(0)	**<0.01**
45–54.9 mm, *n* (%)	14	(26.9)	0	(0)	14	(48.3)	
55–64.9 mm, *n* (%)	9	(17.3)	0	(0)	9	(31.0)	
65–74.9 mm, *n* (%)	3	(5.8)	0	(0)	3	(10.4)	
>75 mm, *n* (%)	3	(5.8)	0	(0)	3	(10.4)	
Intima thickness, µm (range)	60.7	(15–115)	81.7	(20–115)	44.7	(15–100)	**<0.01**

## Data Availability

The data that support the findings of this study are available from the corresponding author upon reasonable request.

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
