# Peer review of "The Role of Telocytes and Telocyte-Derived Exosomes in the Development of Thoracic Aortic Aneurysm"

_ijms, 2022, doi:10.3390/ijms23094730_

Round 1
Reviewer 1 Report
In this paper, the authors explore the possibility that thoracic aortic aneurysm (TAA) is contributed by telocytes and telocyte-derived exosomes. The authors claim that the study ‘supports the TCs involvement in TAA, and SMC phenotype switching by releasing extracellular vesicles’.
What relative contribution to aortic ECM remodelling is due to oxidative stress and what to phenotype switching (caused by TCs?) as the authors propose? This should be discussed.
The paper is in general well written and material is well organized. However, I have some issues with some of the figures.
Fig 1 images can be improved. TC should be labeled red and green (Fig 1B) and 1H, but I cant see the CD34 labeling, at least with the quality of my image (eg in 1H, M, upper panel).
Both HTA/TAA exosomes increase collagen-I (Fig. 3E). Does this not contradict the authors hypothesis?
I fail to see a ‘piriform cell shape’ in Fig 1 F.
line 47 – vascular tone of arterial vessel tone?
Line 69-70 needs rewriting
Table 1. Third column onwards are too narrow and decimals fall on another line below, which is quite confusing.
Arrows in Fig 1 and 2 are not arrows; they are equilateral triangles, so it is not clear where they point to.
Fig 2 G, particle diameter should be (nm)
Fig 2 legend: asterix -> asterisc
narrows -> arrows
line 329 – sentence makes no sense – are equally in vimentin? Please rewrite
lines 171 to 173 – sentences should be opposite: HTA-TC have higher SMA than TAA-TC , not lower, and have lower vomentin and KLF-4 than TAA-TC.
line 160 – please rewrite sentence
line 190: (Fig. 2J), whereas miR levels....instead of . Whereby,..
Author Response
Dear Reviewer,
I am especially happy about your great praise and, of course, that you have taken the appropriate time to elaborate our work in such detail and have given really meaningful, important and good tips.
MANY THANKS!
Reviewer 1:
Comments and Suggestions for Authors
In this paper, the authors explore the possibility that thoracic aortic aneurysm (TAA) is contributed by telocytes and telocyte-derived exosomes. The authors claim that the study ‘supports the TCs involvement in TAA, and SMC phenotype switching by releasing extracellular vesicles’.
What relative contribution to aortic ECM remodelling is due to oxidative stress and what to phenotype switching (caused by TCs?) as the authors propose? This should be discussed.
A: Thank you very much for the very helpful advice. We have now included the topic in our discussion. Line 390-400
The paper is in general well written and material is well organized. However, I have some issues with some of the figures.
Fig 1 images can be improved. TC should be labeled red and green (Fig 1B) and 1H, but I cant see the CD34 labeling, at least with the quality of my image (eg in 1H, M, upper panel).
A: Thank you for the comment. Originally, we got enormously 'into' the construction and presentation of IC staining. Because of the previous work we published on TCs in aortic tissue, in which we dealt excessively with all the markers found in various TCs. Therefore, in this work we will stain only the ost significant markers of aortic TCs, and present the images as simply as possible.HTA show very few TCs, however CD34, a common marker of pericytes in aortas, are naturally represented with a high number. Therefore, the focus is only on the double-positive stained cells shown in the right images (red+green overlay yellow).
Thank you very much!
Both HTA/TAA exosomes increase collagen-I (Fig. 3E). Does this not contradict the authors hypothesis?
A: This is a very good point that, like some other results, still leaves us pondering a bit. Our hypothesis is that the number of exosomes is the determining factor. If we assume that the exosomes of the TCs play a role in remodeling, then an excessive expression of the exosomes would also lead to an excessive collagen production. Note that the number of TCs also changed significantly, not the expression of the exosomes. In the course of our new study, which includes abdominal aortic aneurysms and inflammaging, we are working on how different triggers can specifically influence the expression.
I fail to see a ‘piriform cell shape’ in Fig 1 F.
A: Thanks for the tip. At the last second before the submittals, we improved the Figures, and we must have made the mistake. We have removed the text.
line 47 – vascular tone of arterial vessel tone?
A: Thanks a lot. We have changed the text.
Line 69-70 needs rewriting:
A: Thanks. We have changed the text: Most widely described expression markers for TCs include CD34, Vimentin, ckit and platelet-derived-growth-factor receptor- (PDGFR-). Could be found: Line 72-73
Table 1. Third column onwards are too narrow and decimals fall on another line below, which is quite confusing.
A: Apologies, a formatting error when inserting the table into the template. Should be clearer now. Thanks
Arrows in Fig 1 and 2 are not arrows; they are equilateral triangles, so it is not clear where they point to.
Fig 2 G, particle diameter should be (nm)
Fig 2 legend: asterix -> asterisc
narrows -> arrows
A: Thanks a lot! Corrections have been done.
line 329 – sentence makes no sense – are equally in vimentin? Please rewrite
A: We have changed the text (line 386). Should be more understandable now. Thanks.
lines 171 to 173 – sentences should be opposite: HTA-TC have higher SMA than TAA-TC , not lower, and have lower vomentin and KLF-4 than TAA-TC.
A: thanks you very much! Changes could be found: line 209
line 160 – please rewrite sentence
line 190: (Fig. 2J), whereas miR levels....instead of . Whereby,..
A: Thanks!!! You can find changes line line 227
Reviewer 2 Report
The manuscript titled “The role of telocytes and telocyte-derived exosomes in the development of thoracic aortic aneurysm” reports interesting pilot data, that may be considered for developing future studies with a greater sample size, on biological samples from 29 patients with thoracic aortic aneurysms demonstrating that telocytes are involved in development of this typical cardiovascular disease.
Overall, I feel this manuscript is of high quality, written quite clearly and appropriate for “International Journal of Molecular Sciences”. In my opinion, the manuscript is written in good English (but I am not a native speaker, unfortunately).
This manuscript is basically publishable as is after a change that I suggest.
Introduction:
explain the concept of podomere
Author Response
Dear Reviewer,
I am especially happy about your great praise and, of course, that you have taken the appropriate time to elaborate our work in such detail and have given really meaningful, important and good tips.
MANY THANKS!
Reviewer 2:
Comments and Suggestions for Authors
The manuscript titled “The role of telocytes and telocyte-derived exosomes in the development of thoracic aortic aneurysm” reports interesting pilot data, that may be considered for developing future studies with a greater sample size, on biological samples from 29 patients with thoracic aortic aneurysms demonstrating that telocytes are involved in development of this typical cardiovascular disease.
Overall, I feel this manuscript is of high quality, written quite clearly and appropriate for “International Journal of Molecular Sciences”. In my opinion, the manuscript is written in good English (but I am not a native speaker, unfortunately).
This manuscript is basically publishable as is after a change that I suggest.
Introduction:
explain the concept of podomere
A: Thank you very much. Good point. New text was included.
Reviewer 3 Report
The present manuscript by Aschacher et al. describes the roles of telocytes and of exosomes derived from these cells in the process of aneurysm formation, in particular, the effects on smooth muscle cells. The information has been written clearly and the data have been supported by extensive statistics and they are convincing.
The following minor issues should be taken into consideration before publication of this manuscript.
Abstract:
No remarks. Maybe no need for abbreviation in this section of epithelial progenitor cells in order to limit the number of abbreviated terms. Is it necessary to write telocytes in italics?
Introduction:
Line 47, maybe consider rewriting the phrase on vascular tone (vascular tone of arterial vessel tone). It confuses.
Lines 55 to 59, for readability, consider putting references together (14-27). It makes sense for the readers to know exactly in which references the TCs in the vascular system are found, but not for the other tissues. You can rephrase to ‘and in other major organ systems and tissues’.
Line 69, consider rewriting the first part of the sentence ‘Most TCs described expression markers…’. The meaning of the sentence is not clear in this way. For instance: Most widely described expression markers for TCs include …
Line 77, supra-vasa; a quite recently described cell type; would it be wise to explain this cell type for the reader?
Line 75 and further. The focus is on CD34 cells; why are the vimentin and kit+ cells not described a bit more here? Is there a good reason to emphasize particularly CD34 compared to the other proteins?
Materials and methods:
Line 437, typo ‘osmimum tetroxide’
Results:
Line 106, table 1: consider to give the table more space, in order to have the data (whole numbers and decimals) on the same row. For instance, the number 52.2 is on two rows, making it somewhat difficult to interpret the data. The same applies to table 2 for the words.
Line 106, table 1: the female patients are mentioned in particular. Is there a special, unmentioned, reason for that?
Line 106, table 1: isn’t it remarkable that the ejection fraction is low in a great number of patients that do demonstrate a healthy thoracic aorta?
Typo line 103 ‘Tables 2 and 3 show’ (grammar issue)
Figure 3, the cell migration assay is a very convincing additional experiment.
Discussion:
The hypothesis was partially focused on the importance of numbers of telocytes. It was also shown in the results section that numbers do matter. Why not refer to that in the discussion or conclusion part?
Minor issue: ‘Our aim is to show that…’. I would write this sentence in past tense. The next part ‘For the first part, we classify..’ is fine in the present tense, as it emphasizes the importance of the study.
Author Response
Dear Reviewer,
I am especially happy about your great praise and, of course, that you have taken the appropriate time to elaborate our work in such detail and have given really meaningful, important and good tips.
MANY THANKS!
Reviewer 3:
Comments and Suggestions for Authors
R: The present manuscript by Aschacher et al. describes the roles of telocytes and of exosomes derived from these cells in the process of aneurysm formation, in particular, the effects on smooth muscle cells. The information has been written clearly and the data have been supported by extensive statistics and they are convincing.
A: Thank you very much!
The following minor issues should be taken into consideration before publication of this manuscript.
Abstract:
No remarks. Maybe no need for abbreviation in this section of epithelial progenitor cells in order to limit the number of abbreviated terms. Is it necessary to write telocytes in italics?
A: Thanks for the idea, gladly we made the two corrections.
Introduction:
Line 47, maybe consider rewriting the phrase on vascular tone (vascular tone of arterial vessel tone). It confuses.
A: Good point. Has been corrected.
Lines 55 to 59, for readability, consider putting references together (14-27). It makes sense for the readers to know exactly in which references the TCs in the vascular system are found, but not for the other tissues. You can rephrase to ‘and in other major organ systems and tissues’.
A: Thank you very much. Has been corrected.
Line 69, consider rewriting the first part of the sentence ‘Most TCs described expression markers…’. The meaning of the sentence is not clear in this way. For instance: Most widely described expression markers for TCs include …
A: Thanks. Has been corrected.
Line 77, supra-vasa; a quite recently described cell type; would it be wise to explain this cell type for the reader?
A: I have to agree with them, or rather I have reconsidered. It is neither intellectually significant, nor do I deal with it specifically in the following text, so that I have removed the supra vasa. The main focus is on the well researched pericytes
Line 75 and further. The focus is on CD34 cells; why are the vimentin and kit+ cells not described a bit more here? Is there a good reason to emphasize particularly CD34 compared to the other proteins?
In addition, it should be a discreet hint to suggest the interesting, presumed context between CD34 and the EPC. Because to go into detail later, it would become too confusing and raise too many open questions. Considering that the EPCs differed further and depending on their function in vimentin and ckit expressions. The exact maturation process of EPCs and detailed vim and ckit expression, will be very well discussed in the new study, because in Inflammaging they can play an enormous role - even be used as a marker.
Materials and methods:
Line 437, typo ‘osmimum tetroxide’
A: Corrected. Thanks.
Results:
Line 106, table 1: consider to give the table more space, in order to have the data (whole numbers and decimals) on the same row. For instance, the number 52.2 is on two rows, making it somewhat difficult to interpret the data. The same applies to table 2 for the words.
A: Thank you very much. The formatting has been changed.
Line 106, table 1: the female patients are mentioned in particular. Is there a special, unmentioned, reason for that?
A: The female population is one of the risk factors in the development of aortic aneurysms, so we wanted to present it separately here to show that it does not make a significant difference in this case.
Line 106, table 1: isn’t it remarkable that the ejection fraction is low in a great number of patients that do demonstrate a healthy thoracic aorta?
A: Good point! Part of our control group HTA are aortas obtained from patients undergoing heart transplantation. The disease leading to the heart transplantation is not related to portal vascular disease. This explains the low EF. “A significance was also observed in chronic renal failure, ejection fraction, and a higher proportion of patients taking aspirin and b-blocker, whereby the significance is due to obtained HTA specimens from heart transplant recipients and their chronic heart insufficiency.”
Typo line 103 ‘Tables 2 and 3 show’ (grammar issue)
A: THANKS!
Figure 3, the cell migration assay is a very convincing additional experiment.
A: Yes that's right. We tried hard to place 5 different Telocytes samples and vSMCs on the filter and let them grow, but without success. The application of Matrigel gave 7 different results in 7 different trials. Due to the complexity of cell isolations of TCs from patient’s aortas coming from transplantations (sometimes 6 weeks waiting time), we gave up in frustration and dropped the assay from the experiments.
Discussion:
The hypothesis was partially focused on the importance of numbers of telocytes. It was also shown in the results section that numbers do matter. Why not refer to that in the discussion or conclusion part?
A: Text is included. Thank you very much!
Minor issue: ‘Our aim is to show that…’. I would write this sentence in past tense. The next part ‘For the first part, we classify..’ is fine in the present tense, as it emphasizes the importance of the study.
A: THANKS!